



# Extracting microphysical fault friction parameters from laboratory- and field injection experiments

Martijn Peter Anton van den Ende[1], Marco Maria Scuderi[2], Frédéric Cappa[1,3], and Jean-Paul Ampuero[1]

[1]Université Côte d'Azur, IRD, CNRS, Observatoire de la Côte d'Azur, Géoazur, France
[2]Dipartimento di Scienze della Terra, La Sapienza Università di Roma, Rome, Italy
[3]Institut Universitaire de France, Paris, France

**Correspondence:** M. van den Ende (martijn.vandenende@geoazur.unice.fr)

**Abstract.** Human subsurface activities induce significant hazard by (re-)activating slip on faults, which are ubiquitous in geological reservoirs. Laboratory and field (decametric-scale) fluid injection experiments provide insights into the response of faults subjected to fluid pressure perturbations, but assessing the long-term stability of fault slip remains challenging. Numerical models offer means to investigate a range of fluid injection scenarios and fault zone complexities, and require frictional parameters (and their uncertainties) constrained by experiments as an input. In this contribution, we propose a robust approach to extract relevant microphysical parameters that govern the deformation behaviour of laboratory samples. We apply this Bayesian approach to the fluid injection experiment of Cappa et al. (2019), and examine the uncertainties and trade-offs between parameters. We then continue to analyse the field injection experiment reported by Cappa et al. (2019), from which we conclude that the fault-normal displacement is much larger than expected from the adopted microphysical model (the *Chen-Niemeijer-Spiers* model), indicating that fault structure and poro-elastic effects dominate the observed signal. This demonstrates the importance of using a microphysical model with physically meaningful constitutive parameters, as it clearly delineates scenarios where additional mechanisms need to be considered.

## 1 Introduction

Induced seismicity is of primary concern in human subsurface activities, including geothermal energy production, wastewater and $CO_2$ injection, and hydrocarbon extraction (Ellsworth, 2013). Seismicity triggered around injection sites is generally attributed to elevated pore fluid pressures, which lower the clamping stress that keeps the fault locked (Elsworth et al., 2016). Additionally, recent field injection tests at a decametric scale reveal the importance of aseismic creep in driving seismicity (Duboeuf et al., 2017), and long-range poroelastic effects and earthquake interactions have been inferred to trigger seismicity well beyond the extent of the stimulated region (Catalli et al., 2016; Goebel and Brodsky, 2018; Schoenball and Ellsworth, 2017). To better assess the earthquake hazard associated with the injection and extraction of geofluids, potential mechanisms underlying the nucleation of induced seismic events need to be identified.

Laboratory experiments provide the means to investigate the mechanisms for (unstable) fault slip at high resolution under well-controlled conditions (e.g. Kaproth et al., 2016; Scuderi et al., 2016, 2017; Tenthorey et al., 2003). Many laboratory studies report their results in terms of *rate-and-state friction* (RSF; Dieterich, 1979; Ruina, 1983) parameters, which may serve





as input for numerical modelling studies (Cubas et al., 2015; Kroll et al., 2017; McClure and Horne, 2011; Noda et al., 2017). Unfortunately, it is well established that the RSF parameters depend on a plethora of thermodynamic conditions (Blanpied et al., 1998; Boulton et al., 2019; Chester, 1994; He et al., 2016; Hunfeld et al., 2017), including fluid pressure (Cappa et al., 2019; Sawai et al., 2016; Scuderi et al., 2016), which needs to be accounted for when attempting to extrapolate laboratory measurements to nature through RSF-based numerical models. The relationships between RSF parameters and observable

quantities (such as porosity, grain size, or fluid chemistry) are not well understood, and so great care must be taken to generalise laboratory results to natural systems.

As an alternative approach, decametric-scale fluid injection tests allow one to probe the response of a tectonic fault to fluid pressure perturbations under *in-situ* conditions (Derode et al., 2015; Duboeuf et al., 2017; Guglielmi et al., 2015; Rivet et al., 2016). While these tests provide more direct insights into the (potentially seismic) behaviour of the fault, they are also more

complicated to interpret owing to the complexity inherent to natural faults. Generalisation of the results and extrapolation to other fault or reservoir conditions is therefore challenging. Moreover, fluid injection rates and volumes are limited by regulatory restrictions, which inhibits a comparison with systems characterised by larger injection volumes and rates. Numerical models remain essential to investigate faults in this context (e.g. Dempsey and Riffault, 2019; Rutqvist et al., 2007; Wynants-Morel et al., 2020), which in turn rely on constraints offered by laboratory experiments.

In the present study, we re-interpret the laboratory and decametric-scale fluid injection experiments reported by Cappa et al. (2019) in the framework of the *Chen-Niemeijer-Spiers* (CNS) microphysical model (Chen and Spiers, 2016; Niemeijer and Spiers, 2007). To this end, we propose a robust approach for the extraction of the CNS microphysical parameters from laboratory or field observations based on the relation between fault dilatancy and shear slip, and the temporal evolution of the slip rate. In this Bayesian approach, we examine the uncertainties associated with each parameter, and the trade-offs between

parameters, which are both important for choosing suitable parameter ranges for numerical modelling efforts. Lastly, we discuss the limitations of, and perspectives offered by the adopted microphysical model in the context of induced seismicity modelling.

## 2  Methods

### 2.1  The *Chen-Niemeijer-Spiers* model

To describe the observed laboratory observations of Cappa et al. (2019) in terms of micro-physical quantities, we adopt the

*Chen-Niemeijer-Spiers* (CNS) model proposed by Niemeijer and Spiers (2007) and extended by Chen and Spiers (2016). In the following section, we briefly summarise the basic mechanics of this microphysical model, and the numerical implementation adopted in this study. For a detailed derivation and discussion of this model, we refer to the original works of Niemeijer and Spiers (2007) and Chen and Spiers (2016) (see also Verberne et al., 2020, this issue).

Firstly, the CNS model considers a representative elementary volume of fault gouge of thickness $L$ and porosity $\phi$, which is

subjected to an effective normal stress $\sigma_e$ (i.e. total normal stress minus the fluid pressure) and shear stress $\tau$. In response to this state of stress, the gouge deforms internally through parallel operation of dilatant granular flow and one or more non-dilatant creep mechanisms. The time-scales considered in the present study are too short (of the order of seconds to minutes) to justify





a detailed consideration of the non-dilatant creep component, and hence we focus purely on the granular flow component. As
will be shown later, this simplification is well-warranted by the laboratory observations. In line with this assumption, the shear-
and volumetric deformation of the fault gouge can be described as:

$$\frac{\mathrm{d}\delta}{\mathrm{d}t} = V = L\dot{\gamma}_{gr} \tag{1a}$$

$$\frac{\mathrm{d}\phi}{\mathrm{d}t} = -(1-\phi)\dot{\varepsilon}_{gr} = \tan\psi\,(1-\phi)\,\dot{\gamma}_{gr} \tag{1b}$$

Here, $V$ denotes the rate of slip on the fault $\delta$, and $\dot{\gamma}_{gr}$ and $\dot{\varepsilon}_{gr}$ the shear- and volumetric strain rate of granular flow, respectively
(compression defined positive). We consider only fault-normal volumetric strains (i.e. no fault-parallel expansion/contraction).
The amount of volumetric deformation associated with an increment of shear strain is described by the dilatancy angle $\tan\psi$,
i.e. $\mathrm{d}\varepsilon_{gr} = -\tan\psi\,\mathrm{d}\gamma_{gr}$, and is given by (Niemeijer and Spiers, 2007):

$$\tan\psi = 2H\,(\phi_c - \phi) \tag{2}$$

where $H$ is a geometric constant of order 1, and $\phi_c$ is referred to as the "critical state" porosity, i.e. the maximum attainable
porosity of the gouge. The parameter $H$ represents how much dilatancy is involved when grains are sliding past one another,
and is likely affected by grain shape, angularity, and size distribution. Based on a first-order geometric analysis, Niemeijer
and Spiers (2007) estimated that the maximum dilatancy angle at zero porosity is $\tan\psi = \sqrt{3}$, which puts an upper bound on
$H < \sqrt{3}/2\phi_c$.

The rate of granular flow is itself a function of stress and porosity, and can be written as (Chen and Spiers, 2016):

$$\dot{\gamma}_{gr} = \dot{\gamma}_{gr}^* \exp\left(\frac{\tau\left[1 - \tilde{\mu}^* \tan\psi\right] - \sigma_e\left[\tilde{\mu}^* + \tan\psi\right]}{\tilde{a}\left[\sigma_e + \tau\tan\psi\right]}\right) \tag{3}$$

The reference grain boundary friction coefficient $\tilde{\mu}^*$ corresponds with a shear strain rate $\dot{\gamma}_{gr}^*$, and $\tilde{a}$ is a proportionality constant
for the logarithmic velocity dependence of the grain boundary friction $\tilde{\mu}$, given by:

$$\tilde{\mu} = \tilde{\mu}^* + \tilde{a}\ln\left(\frac{\dot{\gamma}_{gr}}{\dot{\gamma}_{gr}^*}\right) \tag{4}$$

We highlight that $\dot{\gamma}_{gr}$ is exponentially sensitive to the fluid pressure $p$ through the effective stress $\sigma_e = \sigma - p$, and so the CNS
model predicts an acceleration of $V$ upon an increase in the fluid pressure. Moreover, the experiments analysed in this study
are conducted at constant shear stress, so that a force balance (which typically takes the place of Eq. (1a)) is not required.

In the present study, we treat the laboratory sample as a single degree-of-freedom (spring-block) system, with uniform
porosity and internal state of stress. This implies that the fluid pressure is considered to be uniform and constant throughout
the sample, with no coupling between volumetric deformation and fluid pressure. This assumption is valid for samples with
sufficiently high permeability, such that the characteristic time scale of fluid diffusion is smaller than the time scale of defor-
mation. In other words, the sample is assumed to be in equilibrium with the externally applied fluid pressure ("drained") at
all times. In the laboratory experiments of Cappa et al. (2019), the gouge permeability was estimated to be above the intrinsic
permeability of the apparatus ($10^{-14}$ m$^2$), so the sample can be considered to be drained. For low-permeability gouges, such
as shales (Scuderi and Collettini, 2018), coupling between volumetric deformation and fluid pressure needs to be considered
(e.g. Segall and Rice, 1995).



## 2.2 Microphysical parameter inversion procedure


In the simplified CNS framework laid out above, the dynamics of the system are fully governed by $L$, $H$, $\phi_c$, $\tilde{a}$, and $\tilde{\mu}^*$ (which simultaneously constrains $\dot{\gamma}_{gr}^*$), for a given state of stress and initial porosity. In principle, the forward model given by Eq. (1) can be solved iteratively and used to invert laboratory measurements for these constitutive parameters. However, owing to the exponential sensitivity of $V$ to $\phi$ through $\dot{\gamma}_{gr}$, such inversion procedure is unstable and ill-posed. As and alternative, we

propose a two-step inversion procedure that robustly constrains the constitutive parameters. Firstly, we rewrite Eq. (1b) as:

$$\mathrm{d}\phi = \frac{2H}{L}\left(\phi_c - \phi\right)\left(1 - \phi\right)\mathrm{d}\delta \tag{5}$$

where $\mathrm{d}\delta = V\mathrm{d}t$ is an increment of slip across the fault. By integrating the above relation from the initial porosity $\phi_0$ up to $\phi$ (c.f. van den Ende et al., 2018), and recognising that $\Delta L/L = \left(\phi - \phi_0\right)/\left(1 - \phi\right)$, we obtain an expression for the dilatancy $\Delta L$ as a function of slip $\delta$:

$$\frac{\Delta L}{L} = \frac{\phi_c - \phi_0}{1 - \phi_c}\left[1 - \exp\left(-2H\frac{\delta}{L}\left[1 - \phi_c\right]\right)\right] \tag{6}$$

This expression already provides sufficient means to constrain the constitutive parameters $L$, $H$, $\phi_c$, and the initial condition $\phi_0$ without numerically solving the full forward model given by Eq. (1). The second step of the inversion involves constraining the remaining parameters $\tilde{a}$ and $\tilde{\mu}^*$ by comparing Eq. (1a) with the laboratory measured slip rate. Since the slip rate can span orders of magnitude, we perform the inversion in terms of $\ln\left(\dot{\gamma}_{gr}\right)$ (and correspondingly $\ln\left(V\right)$ measured during the

experiment), which renders a more stable inversion task.

Since the proposed inversion protocol does not involve numerically solving a forward model, a single evaluation of either Eq. (6) or (3) yields a sample of the posterior distribution, hence permitting extensive random sampling. To inspect the trade-offs between parameter values and their uncertainties, we cast the protocol above in a Bayesian inversion procedure, in which we estimate the posterior distributions $P\left(m = \{L, H, \phi_0, \phi_c\}|\delta, \Delta L\right)$ and $P\left(\tilde{a}, \tilde{\mu}^*|\delta, V, \sigma_e, \tau, m\right)$ separately. We assume

a uniform prior distribution over a bounded range of admissible parameter values, and a Gaussian likelihood with an unknown data variance $\nu^2$ that is simply treated as a nuisance parameter and co-inverted. The posterior distributions are sampled using an Affine Invariant Markov Chain Monte Carlo ensemble sampler as implemented in the Python *emcee* package (Foreman-Mackey et al., 2013). While it is also possible to estimate the posterior distributions from numerically solving the forward problem, each forward model evaluation from $t = 0$ up to the point where $V > 1\,\mathrm{mm\,s^{-1}}$ takes several tens of seconds on a

single CPU. The practical reason for this is that the fault is critically stressed, and hence requires small time-step evaluations to ensure sufficient numerical accuracy and stability.

## 3 Analysis of fluid-injection tests of Cappa et al. (2019)

### 3.1 Laboratory experiment

We apply the above procedure to the laboratory fluid injection experiment performed by Cappa et al. (2019) – see Fig. 1.

In this experiment, a carbonate gouge sample was subjected to a constant shear stress of $\tau = 1.2\,\mathrm{MPa}$ and a total normal





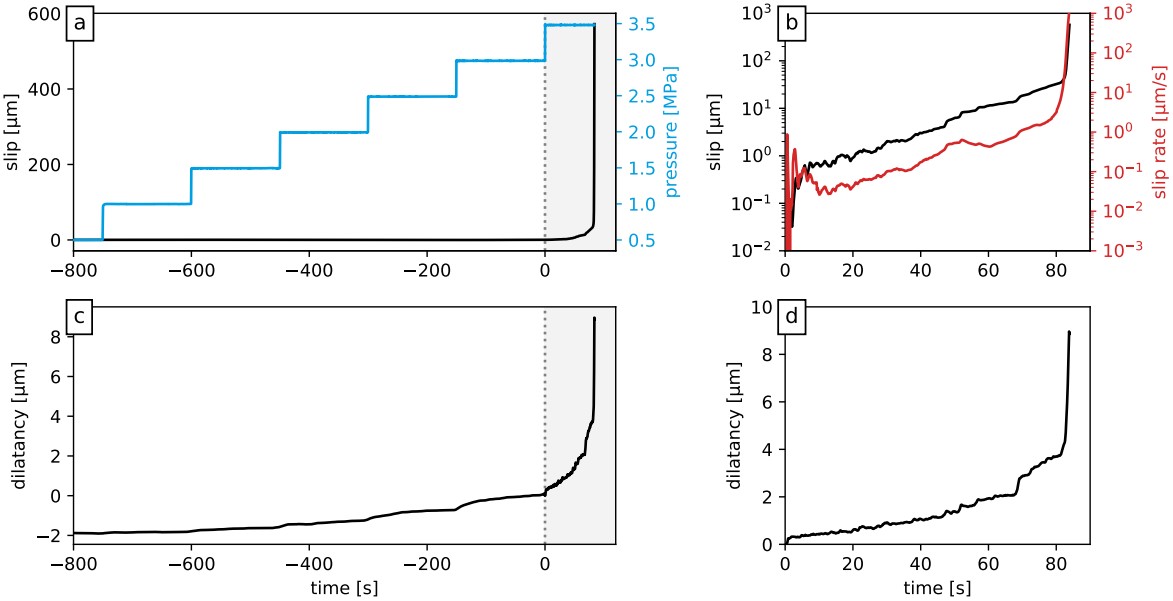

**Figure 1.** Overview of laboratory measurements of Cappa et al. (2019). The fault slip and dilatancy recorded over the full creep stage of the experiment are shown in panels a) and c), respectively, along with the fluid pressure for reference. The final stage of the experiment (grey-shaded area of panels a) and c)) is enlarged in panels b) and d).

stress of $\sigma = 5\,\mathrm{MPa}$. The fluid pressure was step-wise increased every 150 s with steps of 0.5 MPa, until the sample 'failed' macroscopically at a fluid pressure of $p = 3.5\,\mathrm{MPa}$. Prior to the final stage of pressurisation, only negligible amounts of slip were measured, and hence we focus our inversion efforts on the final stage of the experiment in which the sample measurably accelerated. Additionally, through the stage of fluid injection, no gouge compaction was measured, supporting our assumption

made prior to Eq. (1) that the time-dependent creep rate is negligible compared to the rate of granular flow.

We first fit Eq. (6) to the measured dilatancy as a function of slip. Since the data are sampled uniformly in time but not in slip (as the sample deformation is accelerating), we interpolate the slip data to assign uniform weight to each measurement during the inversion. The bounds on the prior distribution are given by $10 < L < 100\,\mathrm{\mu m}$, $0.1 < H < 1$, $0.1 < \phi_0 < \phi_c$, and $0.2 < \phi_c < 0.4$. The resulting posterior distributions are presented in a corner plot (Fig. 2), showing appreciable trade-offs between

$L$ and $H$, and between $\phi_0$ and $\phi_c$. Nonetheless, the parameters $L$ and $H$ are reasonably well resolved as $L = 64.3 \pm 8.9\,\mathrm{\mu m}$ and $H = 0.45 \pm 0.06$ (median $\pm$ 1 standard deviation). And while $\phi_0$ and $\phi_c$ trade-off almost perfectly and hence span a near-uniform distribution over the permitted parameter range, their difference is well resolved as $\phi_c - \phi_0 = 0.09 \pm 0.01$. These parameter values are perfectly consistent with previous studies (e.g. Chen and Spiers, 2016; van den Ende et al., 2018). Although the inferred layer thickness $L$ is much less than the total thickness of the sample (initially around 5 mm), one should

keep in mind that deformation localises in a much narrower zone, so that the effective thickness of the actively deforming region of the gouge is much less than the total sample thickness. In similar experiments conducted by Scuderi et al. (2017),



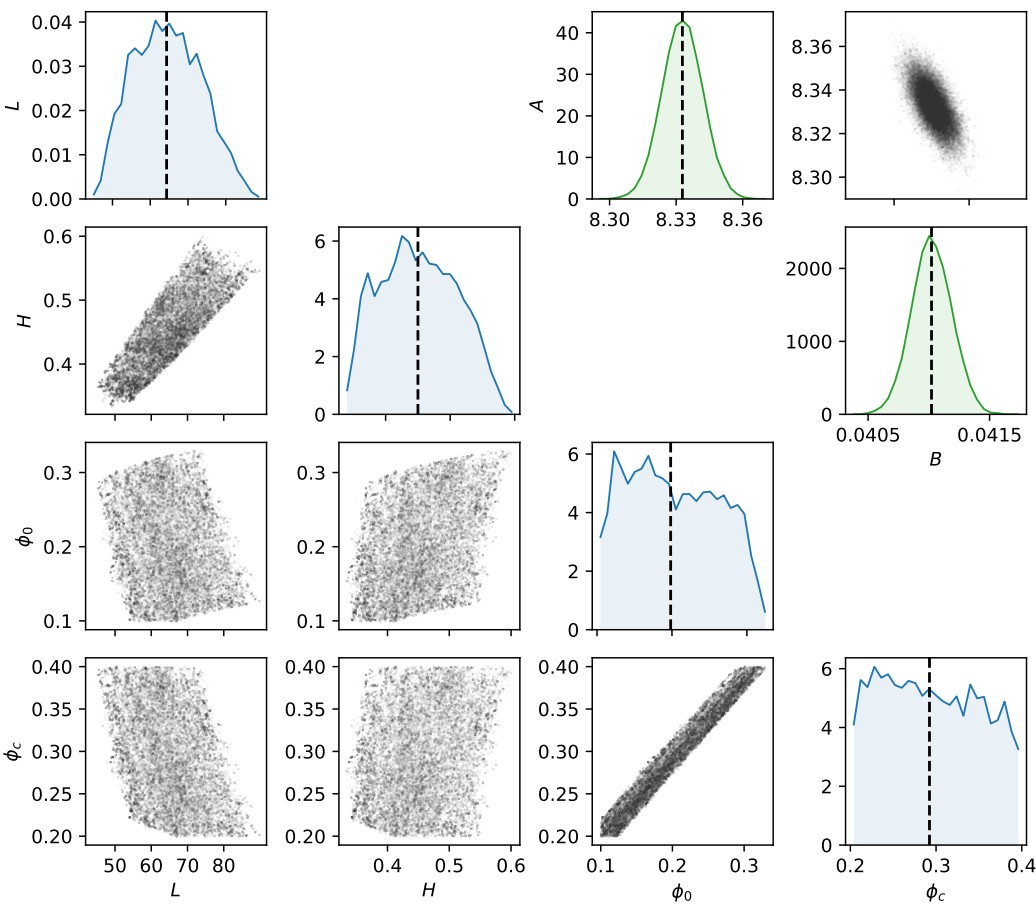

**Figure 2.** Lower triangle (blue): Corner plot of the posterior distributions of the inverted parameters $L$, $H$, $\phi_0$, and $\phi_c$ (marginalised over the nuisance parameter $\nu^2$) for the laboratory injection experiment. The main diagonal panels show the posterior probability density distribution of each parameter, whereas the off-diagonal panels show the co-variance of posterior samples. Upper triangle (green): Corner plot of the posterior distributions of $A$ and $B$ (see main text).

the localised region was observed to have a thickness of 10-20 μm, which was inevitably affected by post-experiment compaction. Hence, our inferred estimate of 64 μm seems appropriate for an actively deforming localised gouge layer. Upon inspection of Eq. (6), we can formulate the mapping between layer thickness and slip as $\Delta L = A\left[1 - \exp\left(-2B\delta\right)\right]$, and infer

$A = L\left(\phi_c - \phi_0\right)/\left(1 - \phi_c\right)$ and $B = H\left(1 - \phi_c\right)/L$ as lumped parameters (upper triangle of Fig. 2). Since $A$ and $B$ are the only parameters directly constrained by the data, the original four parameters depend on them and show strong trade-offs.

We continue by fitting the (logarithm of) measured slip rate based on Eq. (3), using the parameter values inferred in the previous step to compute the time-evolution of $\tan\psi$. Without loss of generality, we define $\dot{\gamma}_{gr}^* = 1\,\mu\text{m s}^{-1}/L$, so that $\tilde{\mu}^*$ represents the grain boundary friction coefficient at a slip rate of $V = 1\,\mu\text{m s}^{-1}$. Since $\phi_0$ and $\phi_c$ individually are ambiguous,





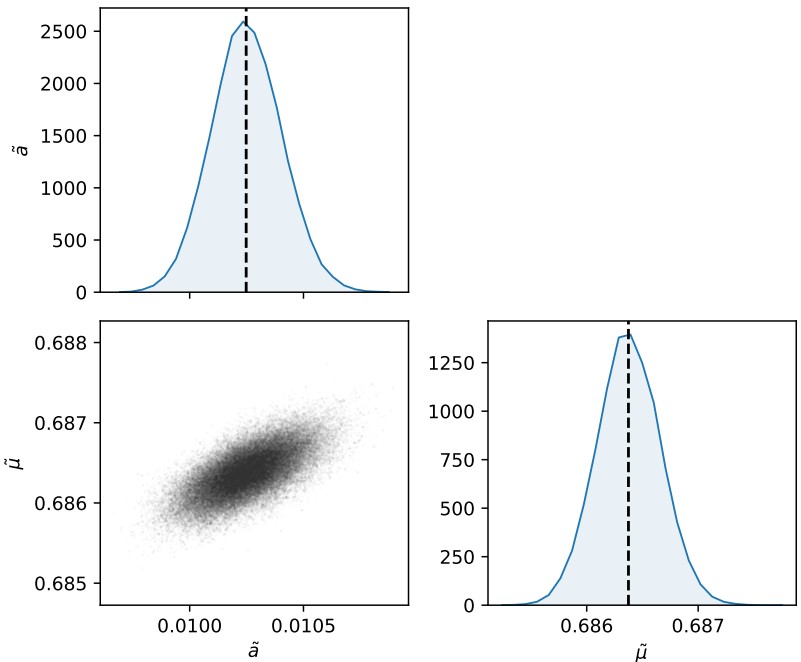

**Figure 3.** Corner plot of the posterior distributions of the inverted parameters $\tilde{a}$ and $\tilde{\mu}$ (marginalised over the nuisance parameter $\nu^2$) for the laboratory injection experiment. The main diagonal panels show the posterior probability density distribution of each parameter, whereas the off-diagonal panel shows the co-variance of posterior samples.

we take $\phi_0 = 0.25$ and increment this value by the inverted $\phi_c - \phi_0$ to obtain $\phi_c = 0.34$. The slip rate parameters are extremely well resolved (see Fig. 3), and found to be $\tilde{a} = (10.26 \pm 0.15) \times 10^{-3}$ and $\tilde{\mu}^* = 0.6852 \pm 0.00028$, with minimal trade-off between the two parameters. With these parameters, the fit to the slip rate data is excellent (Fig. 4b).

     Finally, for verification, we numerically solve the forward model given by Eq. (1) with the parameters obtained in the inversion procedure (Fig. 4c and d). While we obtain an excellent fit with the observed time-evolution of slip and dilatancy, we 150   also find that the forward model is extremely sensitive to the initial condition $\phi_0$. While the overall features of the simulated sample response are similar, the exponential sensitivity to porosity leads to critical behaviour and strong variations in the timing of the sample failure. This is highlighted in Fig. 4c and d, where we vary the initial porosity between 0.2454 and 0.2509 ($-1$ and $+5$ % around the reference value of 0.25). The initial condition that gives the best match in terms of the onset of accelerated slip is 1.3 % above the initially chosen value of $\phi_0 = 0.25$, although we assign no significance to such a tiny deviation.

**3.2   Field experiment**

Encouraged by the results of the proposed inversion method for the laboratory experiment, we continue to apply the same procedure to the field injection test of Cappa et al. (2019). Like in the laboratory experiment, the *in-situ* pressurisation of a tectonic fault triggered accelerating slip, and associated with it fault opening (dilatancy) – see Fig. 5a. While the acceleration of

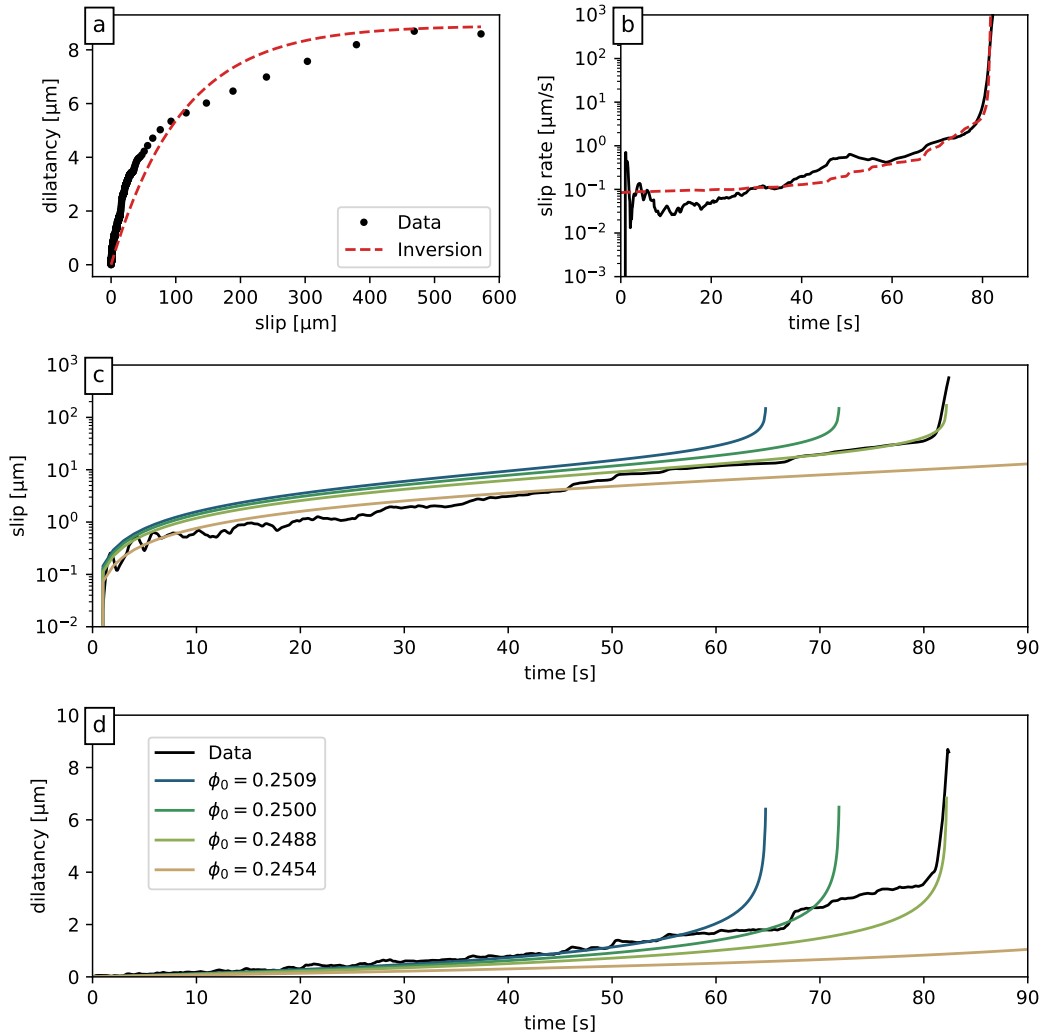

**Figure 4.** Results of the inversion procedure. a) Dilatancy versus slip, with the inversion curve given by Eq. (6); b) Slip rate versus time, with the inversion curve given by Eq. (1a); c) and d) Forward model results of slip and dilatancy for a (narrow) range of initial porosity $\phi_0$ (as indicated in the legend). The reference value of $\phi_0$ is obtained from the inversion of Eq. (6).

shear and normal displacement on the fault was more gradual than in the laboratory experiment, a phase of rapidly accelerating

slip at $t > 800\,\mathrm{s}$ can be clearly seen. The amount of dilatancy measured as a function of slip (Fig. 5b) was proportionally more than in the laboratory experiment by at least one order of magnitude, so we expect a-priori that the frictional parameters inferred from the laboratory cannot immediately describe the behaviour of the fault *in-situ*. Indeed, when we perform the inversion of the dilatancy-slip data from the field experiment, we find median values of $L = 4.9 \pm 1.0\,\mathrm{mm}$, $H = 13.0 \pm 2.5$, and $\phi_c - \phi_0 = 0.082 \pm 0.016$ (Fig. 6).





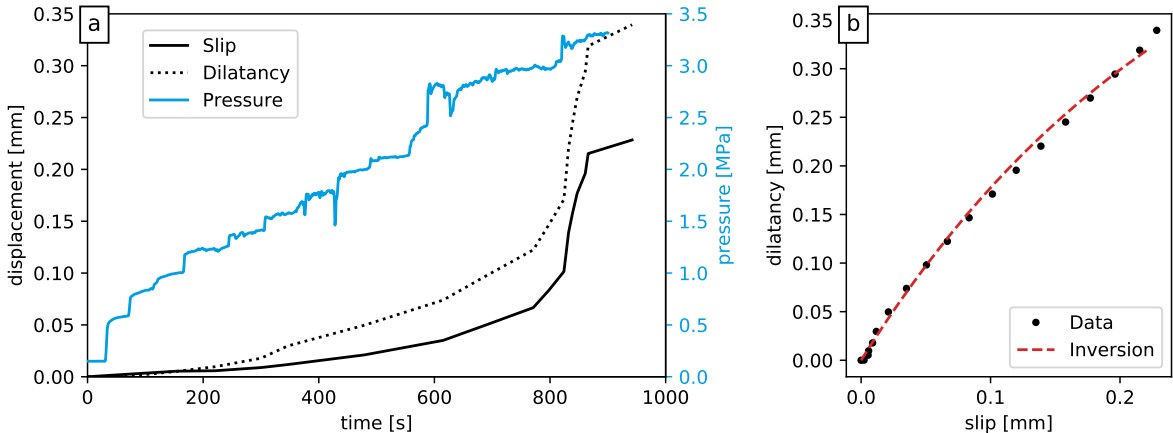

**Figure 5.** a) Measurements of fault shear- and normal displacement, and fluid pressure during the field injection test of Cappa et al. (2019); b) Inversion of the dilatancy measured during the field injection experiment. To produce a reasonable fit to the data, an unrealistically high value of the dilatancy parameter $H = 6.4$ was required.

While these other values seem entirely reasonable, the inferred value of $H$ is well above the estimated upper bound of $\sqrt{3}/2\phi_c \approx 2.9$. This suggests that the CNS model is unable to explain the relationship between fault slip and fault opening in this experiment. In the CNS model, dilatancy is envisioned to originate from grain sliding and rolling, neighbour swapping, and "jostling", which requires a volume increase of the gouge to accommodate. However, the model fault itself is mathematically planar, and so no dilatancy occurs due to geometric constraints. In the case of a macroscopically non-planar fault geometry

(as is inevitable for tectonic faults; Candela et al., 2012), additional dilatancy (with associated permeability changes) at the onset of slip is necessary. Moreover, poro-elastic effects (elastic fault opening) due to fluid pressure changes are not considered here. The inability of the CNS model to describe the fault opening with a reasonable choice of parameters is therefore not a shortcoming of the CNS model (which describes the mechanics of a small representative volume element), but is rather due to an incomplete coupling with processes that transcend the scale envisioned by the CNS model.

Since the CNS model fault strength (and therefore the fault slip rate) is directly controlled by the dilatancy parameter $H$, it is unwarranted to attempt to infer $\tilde{a}$ and $\tilde{\mu}^*$ based on the parameters inferred from the dilatancy. While this may seem like a severe limitation of the CNS model, it actually serves as an important indication of the applicability of the model, and the validity of its parameters, when attempting to extrapolate to nature. Moreover, the basic mechanics of the CNS model are still expected to govern the strength and slip rate of the fault, even though part of the model predictions (the dilatancy) cannot be

constrained by independent measurements. By numerically solving the forward model, the fault slip as a function of time and fluid pressure may be reproduced within a reasonable range of parameter values, for which the predicted fault opening would likely be much less than measured by Cappa et al. (2019).

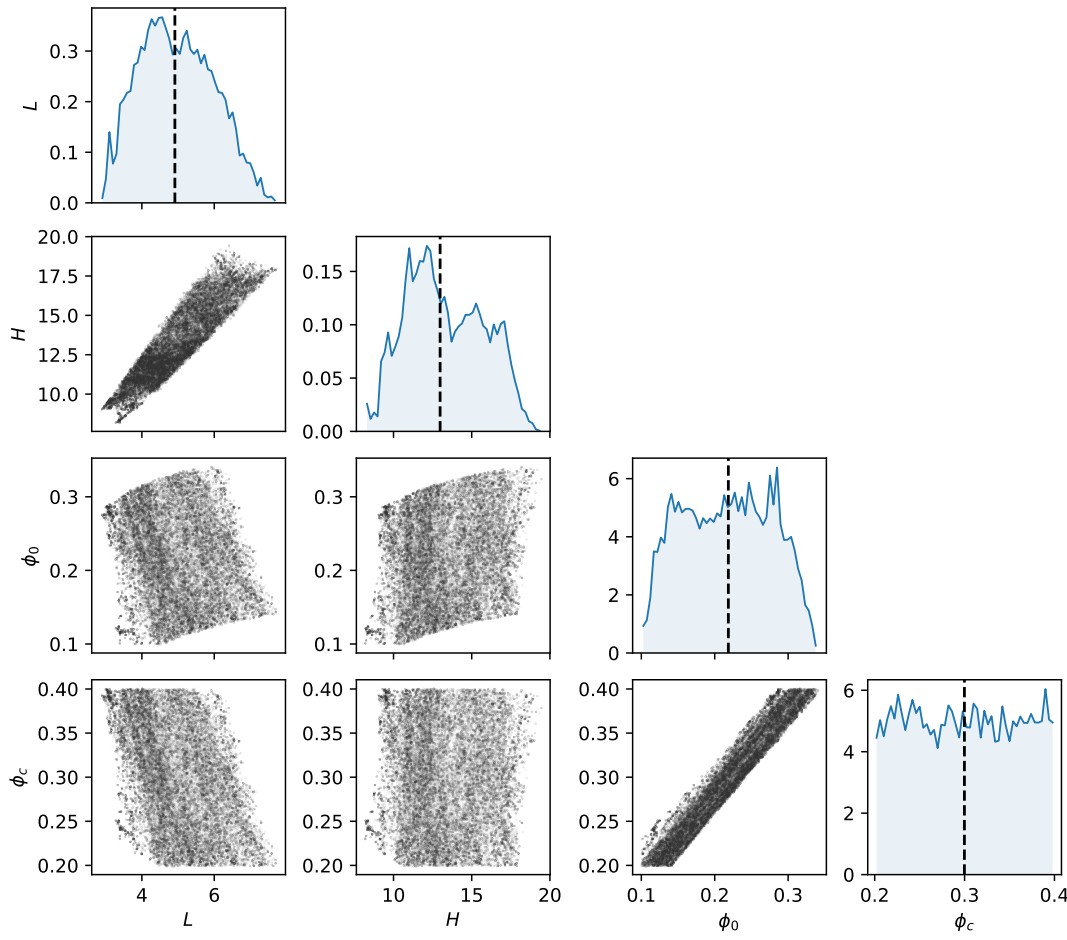

**Figure 6.** Corner plot of the posterior distributions of the inverted parameters $L$, $H$, $\phi_0$, and $\phi_c$ (marginalised over the nuisance parameter $\nu^2$) for the field injection experiment. The main diagonal panels show the posterior probability density distribution of each parameter, whereas the off-diagonal panels show the co-variance of posterior samples.





## 4  Discussion

### 4.1  Comparison with rate-and-state friction

Traditionally, laboratory experiments are interpreted within the framework of rate-and-state friction (RSF), commonly presented as (Dieterich, 1979; Ruina, 1983):

$$\mu(V,\theta) = \mu^* + a\ln\left(\frac{V}{V^*}\right) + b\ln\left(\frac{V^*\theta}{D_c}\right) \tag{7a}$$

$$\frac{\mathrm{d}\theta}{\mathrm{d}t} = \begin{cases} 1 - \frac{V\theta}{D_c}, & \text{``ageing law"} \\ -\frac{V\theta}{D_c}\ln\left(\frac{V\theta}{D_c}\right), & \text{``slip law"} \end{cases} \tag{7b}$$

where $\mu(V,\theta)$ is the macroscopic friction coefficient at slip rate $V$ and "state" $\theta$, $\mu^*$ is a reference friction coefficient at slip

rate $V^*$, and $a$, $b$, and $D_c$ are empirical constants. As has been shown by Chen et al. (2017), the CNS model is asymptotically identical to RSF for small departures from steady-state, for which the CNS equivalents of the RSF parameters $a$, $b$, and $D_c$ can be treated as constants. For large departures from steady-state, the behaviours predicted by CNS and RSF diverge, as the aforementioned parameters can no longer be considered to be constant (van den Ende et al., 2018). Nonetheless, because of their similarity, the limitations of the CNS model also apply to rate-and-state friction. One advantage of using the CNS microphysical

model over traditional RSF, is that the governing parameters have a more physically meaningful interpretation. Even though numerous studies have attempted to elucidate the physical origin of RSF (Aharonov and Scholz, 2018; Brechet and Estrin, 1994; Ikari et al., 2016; Putelat et al., 2011), in practice these theoretical constraints are not considered. Instead, it is more convenient to constrain the RSF parameters empirically through laboratory velocity-step experiments (Blanpied et al., 1998; Carpenter et al., 2016; Chester, 1994; Hunfeld et al., 2017; Reinen and Weeks, 1993). With these laboratory measurements of

the RSF parameters, fault slip observed during decametric-scale fluid injection tests can be accurately modelled (Cappa et al., 2019), although the same behaviour can be obtained for a wide range of parameter values: in the study of Cappa et al. (2019) a similar fit to the data was obtained for velocity-weakening $((a-b) < 0)$ and velocity-strengthening $((a-b) > 0)$ friction, even though seismic slip can only be produced in the former case of velocity-weakening friction. Hence, more observational constraints are required to distinguish between the different types of behaviour.

Aside from the fault-parallel slip, the fault opening potentially provides a second prominent constraint. In the classical RSF framework, volumetric deformation is not explicitly accounted for. Traditionally, the state parameter $\theta$ has been interpreted as encoding the average life time of asperity contacts (at steady-state), or the relative area of asperity contacts (Dieterich, 1994; Scholz, 2019), both of which do not entail volumetric deformation of the fault gouge. Empirical relations between the state parameter $\theta$ and porosity have also been proposed (Segall and Rice, 1995; Sleep, 2005) and used in hydro-mechanical modelling

(Jeanne et al., 2018), but these relations are typically not employed as additional constraints of the RSF constitutive parameters. Moreover, relations between the steady-state coefficient of friction (and its velocity-dependence) have been established based on energy balance considerations (Beeler et al., 1996; Marone et al., 1990). Since these relations pertain to the steady-state coefficient of friction, they do not apply to non-steady state conditions (for which $\mathrm{d}V/\mathrm{d}t \neq 0$ and $\mathrm{d}\phi/\mathrm{d}t \neq 0$) and do not offer





additional insight on the relationship between $\theta$ and $\phi$. On the other hand, volumetric deformation is an integral part of the CNS

model, hence allowing (and requiring) us to incorporate these measurements to arrive at a better constrained set of parameters.

### 4.2    Relationships between experiments and nature

While the CNS microphysical parameters can be directly estimated from laboratory experiments, their incorporation into numerical models of tectonic faults may be subject to moderation based on geological or physical considerations. In laboratory experiments conducted at room ambient conditions and comparatively high deformation rates (of the order of $\mathrm{\mu m\,s^{-1}}$ up to

$\mathrm{mm\,s^{-1}}$), the gouge porosity remains close to the critical state porosity. Likewise, in the laboratory experiment of Cappa et al. (2019), the initial porosity was estimated to be less than 0.1 units of porosity below the critical state porosity. Given longer time-scales and higher temperatures, compaction induced by one or more time-dependent creep mechanisms (such as pressure solution creep or subcritical crack growth) would gradually reduce the porosity of the gouge, thereby increasing its strength and critical fluid pressure at which the fault slip rates become appreciable. In numerical simulations of fault slip, the initial

state of a tectonic fault is likely not the same as for the laboratory fault. Fortunately, this initial state could be estimated from microstructural analyses of drill cores. Moreover, the choice of initial state of the fault does not affect any of the other frictional parameters of the CNS model. This is in contrast to rate-and-state friction, where the initial value of the state parameter ($\theta$ at $t = 0$) should also affect the magnitude of $b$, which has been found to increase with decreasing porosity (or increasing $\theta$; Chen et al., 2015, 2017).

The property that $b$ (or more precisely: $b/D_c$) is sensitive to the gouge porosity can also be derived from stability analysis of the CNS model. Consider the general criterion for unstable slip of a spring-block:

$$\frac{\mathrm{d}\tau}{\mathrm{d}\delta} = \frac{1}{V}\left(\frac{\partial\tau}{\partial\phi}\frac{\mathrm{d}\phi}{\mathrm{d}t} + \frac{\partial\tau}{\partial V}\frac{\mathrm{d}V}{\mathrm{d}t}\right) \leq -K \tag{8}$$

where $K$ is the shear stiffness of the fault. For an instantaneous step-change in velocity, $\mathrm{d}V/\mathrm{d}t = 0$ for $t > 0$. Assuming that unstable slip is governed by the onset of granular flow, the shear strength is given by the CNS model as (Chen and Spiers,

235    2016):

$$\tau = \frac{\tilde{\mu} + \tan\psi}{1 - \tilde{\mu}\tan\psi}\sigma_e \tag{9}$$

Hence, using Eq. (1b), the stability criterion can be expressed in terms of microstructural quantities as (van den Ende et al., 2018):

$$K \leq 2H\left(1 - \phi\right)\tan\psi\,\frac{1 + \tilde{\mu}^2}{\left(1 - \tilde{\mu}\tan\psi\right)^2}\frac{\sigma_e}{L} \tag{10}$$

In the vicinity of steady-state, the above statement should be identical to the stability criterion derived from rate-and-state friction, i.e. (Rubin and Ampuero, 2005):

$$K \leq K_b = \frac{b\sigma_e}{D_c} \tag{11}$$





Here, $K_b$ is a critical stiffness value that facilitates acceleration of slip (seismic or aseismic). From the comparison of the two inequalities, it can be concluded that the $K_b$ therefore must increase with decreasing porosity. This was also observed in the

Discrete Element Model simulations of van den Ende and Niemeijer (2018), which were conducted completely independently of the assumptions and limitations of the CNS model. We note that the comparison between Eq. (10) and (11) only holds in the vicinity of steady-state. Nonetheless, Eq. (10) can be used to describe the stability of fault slip far from steady-state, circumventing the issue of the velocity- and state-dependence of $a$, $b$, and $D_c$ (as observed by Cappa et al., 2019; den Hartog and Spiers, 2013; Reinen et al., 1992; Takahashi et al., 2017, and many others).

Combining now the observations made in Section 3 with the discussion above, we propose that the seismogenic potential of faults subjected to fluid pressure perturbations is best described in terms of the dilatant behaviour of the fault, and its initial degree of compaction. One can infer the microphysical parameters $H$, $\phi_c$, $\tilde{a}$, and $\tilde{\mu}^*$ from laboratory experiments, and assume reasonable *in-situ* values of $\phi$ and $L$ for the tectonic fault to simulate its response to a changing stress field (fluid pressure). If permitted by the numerical method, fault non-planarity, permeability changes, and elastic moduli reduction may be introduced

to add further complexity, as anticipated based on the results of Section 3.2. In this way, the evident pressure- and velocity-dependence of the rate-and-state friction parameters can directly be accounted for in a self-consistent and transparent manner, and the model outcomes interpreted in terms of physical observables.

## 5  Conclusions

In this work, we analysed the fluid injection experiments conducted by Cappa et al. (2019) in the laboratory and *in-situ*, in terms

of the *Chen-Niemeijer-Spiers* (CNS) microphysical model. We proposed a Bayesian inversion approach to extract the governing parameters without the need for numerically solving the forward problem, while elucidating the uncertainties and trade-offs between the model parameters. We showed that while the localised gouge layer thickness $L$ and the dilatancy parameter $H$ can be well resolved, the initial- and critical state porosities trade-off perfectly, so that only their difference $\phi_c - \phi_0$ can be resolved in the experiments. When numerically solving the forward model with the inferred parameter values, we obtained

almost perfect agreement with the measurements, indicating that the CNS model accurately describes fault deformation in response to a fluid pressure perturbation. When the same inversion approach was applied to a decametric-scale field injection experiment, we found that the inferred parameters fell outside of the feasible range of values, highlighting the relevance of other mechanisms, such as fault structure and poro-elastic effects, in this scenario.

     The excellent agreement between the CNS model and the laboratory data allows us to interpret the dynamics of the fault in

terms of volumetric deformation (porosity changes). By doing so, we circumvent the velocity-dependence of the rate-and-state friction parameters $a$, $b$, and $D_c$, which increases the predictive power of numerical models of natural faults. Adopting the CNS model expedites the extrapolation of laboratory results to nature, and permits better assessment of the applicability of the model and accuracy of the parameter values.



*Code and data availability.* A Python script that reproduces the results and figures in this manuscript, along with the laboratory and field
injection data, are available at https://doi.org/10.6084/m9.figshare.12613007.

*Author contributions.* MvdE conceptualised the study and performed the analyses. MS and FC provided the laboratory and field data. JPA
supervised MvdE. All authors discussed and prepared the contents of the manuscript.

*Competing interests.* The authors declare no competing interests

*Acknowledgements.* MvdE, FC, and JPA are supported by French government through the UCA$^{\text{JEDI}}$ Investments in the Future project
managed by the National Research Agency (ANR) with the reference number ANR-15-IDEX-01.





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
