# Peer review of "Extracting microphysical fault friction parameters from laboratoryand field injection experiments"

_Solid Earth, 2020_

## Referee Comment (RC1) · Anonymous Referee #1 · 9 Sep 2020

The manuscript present new insights regarding the inversion of frictional parameters from laboratory and field injection experiments. The authors are using a Bayesian inversion approach, using the evolution of the normal dilatancy with slip to outputs the best solution for geometric and frictional parameters used in CNS model (L, H and phi_0 and phi_c). Then They compute the solution for the slip rate based on the microphysical model. The paper is well written and is of broad interest for the community. I have few comments that I think could improve the clarity of the manuscript.

Major comments

1. Comments regarding the Figure of the paper: The figures are of high quality, but

[Figure]

should be completed following the comments above.

Figure 1: Authors could also present the evolution of the shear stress on one of the plots.

Figure 2: Authors should complete the legends of the axis. The Y axis of the distribution plots looks to refer to a probability function rather than the parameter. Authors should also present the units of each parameter, also because the scale is different between the lab and the field experiments.

Figure 4: Maybe it could be relevant to plot directly the experimental values of dilatancy at a given slip versus the theoretical values to show the trend and the robustness of the prediction. Same could be also done for slip rate as a given time.

Figure 5: Same comment for panel b.

Figure 6: Same comments than for figure 2.

2. The model used here to describe the time evolution of slip at the onset of fault reactivation is based on the properties of the gouge layer. From the micromechanic model, the physical response of the system, and notably an upper bound value for the dilatancy, is expected (in my opinion) to be controlled by the average grain size of the grains. However, in the present study, the grain size is not an important parameter. Can authors comment on it?

3. The results presented here show that a good fit of the experimental observations is obtained using the parameters output from the inversion. However, it looks that the model explains well the increase in slip rate, but does it also explain the decrease in slip velocity observed in the field injection experiments that occur just after the onset of rapid slip (Cappa et al., figure 1)? Or is the dilatancy I am also surprise to see how tight variations in initial porosity can induce such large variations in the slip history presented in figure 4c and 4d. Do authors things that this behavior could be observed in the laboratory?

4. The main observations made here is that the values of H allowing to fit the experimental data recorded during the field injection are strongly larger than the upper bound definition proposed by Niemeijer and Spiers. The model proposed here explain dilation by shear at the contacts of gouge grains. However, along fault interface, dilation is expected to be mostly controlled by fault geometry and long scale roughness encountered within the slip domains. It is stated in the text lines 166-174 but do author think about an adaptation of the model to include a second dilation angle due to fault geometry?

5. Finally, this model is expected to describe the slip behavior of the fault in drained conditions (homogeneous fluid pressure distribution), however, I believe it is not the case in the fluid injection experiments where the slip front outgrowths the fluid pressure front (Bhattacharya and Viesca). In addition, a fluid pressure gradient is expected, even at the scale of the laboratory in partially drained conditions (Passelegue et al., 2018) maybe authors should add a small comment about it in the manuscript.

Minor comments:

Equation 1b: I am probably wrong but I am not sure that the equation described here refer directly to d_phi/d_t. It looks to me more related to the dilatancy rate of the gouge layer.

Lines 98: It is probably not changing a lot, but I wonder if you should not compute Delay_L/L=(phi-phi_0)/(1-/phi_0) to consider the initial fraction of the matrix.

Maybe I missed it but I do not see the definition of a∼ in the text, that should be of the form d\mu∼/d(ln V) or something like that.

Part 4.1: I wonder about the relevance of this part here since the limitation of RSF has been already mentioned in the introduction.

---

## Referee Comment (RC2) · Anonymous Referee #2 · 18 Sep 2020

The manuscript looks to constrain rock frictional properties via laboratory and field experimental data. The authors use the CNS microphysical model for frictional strength and dilatancy evolution and invert. The work stands apart in that (i) it's an effort directed away from more classical models (e.g., rate- and state-dependent friction) and (ii) the careful attention is paid to the parameter inversion itself, including Bayesian inference of parameter probably distributions as well as parameter covariance. The authors are forthcoming in their results, highlighting both the good fit of laboratory data as well as the inability of their model to fit the field experiments with plausible parameters, and the possibility for arbitrary choices for some model parameters (e.g., phi_0 or phi_c). I think the paper is an exemplary model of its kind.

Minor comments:

-In passing from (5) to (6) it seems implicitly assumed that L in the RHS of (5) is constant when integrating, whereas the resulting expression (6) implies that a substantial evolution of L is possible. Could the problem be closed by presenting an equation such as dL/dphi = f(L, phi)?

-Equation (11): wouldn't the linear stability analysis results of Ruina, in which (b-a) takes the place of (b) in (11), provide a more relevant critical stiffness close to steady state?

-Could the authors flesh out more directly why there is such a strong dependence on the initial porosity (e.g., Fig. 4)?

Line 94, typo: "an"

---

## Author Comment (AC1) · 8 Oct 2020

Major comments

**[R1.1]** *Comments regarding the Figure of the paper: The figures are of high quality, but should be completed following the comments above.*

*Figure 1: Authors could also present the evolution of the shear stress on one of the plots.*

As mentioned in line 120 of the original manuscript, the shear stress in the experiment is maintained constant at 1.2 MPa, and is not allowed to evolve. A plot of the shear stress would thus simply be a straight line and therefore redundant for this manuscript. For the shear stress evolution prior to the fluid injection stage (not analysed here), we refer to the original manuscript of Cappa et al. (2019).

**[R1.2]** *Figure 2: Authors should complete the legends of the axis. The Y axis of the distribution plots looks to refer to a probability function rather than the parameter. Authors should also present the units of each parameter, also because the scale is different between the lab and the field experiments.*

*Figure 6: Same comments than for figure 2.*

Figures 2, 3, and 6 have been modified following the reviewer's suggestions. For clarity, the y-axes of the probability distributions have been removed (since the amplitude of the probability density is not directly informative), and the parameter symbols that decorate the rows and columns are now written upright (i.e. vertically). It is mentioned in the figure captions that the main diagonal of each corner plot shows the probability density distribution of a given parameter. Hopefully with these modifications our intentions are clear to the reader, without crowding the panels with too many axis decorations.

**[R1.3]** *Figure 4: Maybe it could be relevant to plot directly the experimental values of dilatancy at a given slip versus the theoretical values to show the trend and the robustness of the prediction. Same could be also done for slip rate as a given time.*

*Figure 5: Same comment for panel b.*

By plotting a predicted quantity against its measured counterpart (the "ground truth", if you will), systematic errors such as a constant or proportional bias can be easily recognised in otherwise uncorrelated data. Fortunately the data considered here are correlated in time, so that systematic errors can also be observed in the time series. The comparison between the model predictions and the measured values of dilatancy and slip rate, as shown in Fig. 4a and 4b of the manuscript, seems to indicate that, even though the fit is not perfect, the model does not systematically over- or underestimate the measured data. For completeness we include the requested figure here, but we believe that it does not contribute to any new insights. Thus, for brevity, we did not incorporate this new figure into the revised manuscript.

[Figure]

[Figure]

**[R1.4]** *The model used here to describe the time evolution of slip at the onset of fault reactivation is based on the properties of the gouge layer. From the micromechanic model, the physical response of the system, and notably an upper bound value for the dilatancy, is expected (in my opinion) to be controlled by the average grain size of the grains. However, in the present study, the grain size is not an important parameter. Can authors comment on it?*

The dilatancy is controlled by the parameters $H$ and $\phi_c$. We agree that these likely depend on grain shape and size distribution, which we briefly mentioned in lines 68-72 of the original manuscript. We now made this statement more explicit as:

"*The parameter $H$ represents how much dilatancy is involved when grains are sliding past one another, and is likely affected by grain shape, angularity, and size distribution. […] Likewise, the critical state porosity $\phi_c$ is likely not a universal constant. Nonetheless, in the absence of tight theoretical constraints on $H$ and $\phi_c$ we treat these quantities as constant parameters.*"

**[R1.5]** *The results presented here show that a good fit of the experimental observations is obtained using the parameters output from the inversion. However, it looks that the model explains well the increase in slip rate, but does it also explain the decrease in slip velocity observed in the field injection experiments that occur just after the onset of rapid slip (Cappa et al., figure 1)? Or is the dilatancy I am also surprise to see how tight variations in initial porosity can induce such large variations in the slip history presented in figure 4c and 4d. Do authors things that this behavior could be observed in the laboratory?*

The field experiment is substantially more complex than the laboratory experiment, so details in the field experiment, such as the deceleration following the onset of rapid slip, are not easily explained either by the laboratory experiments or by any model (see Fig. 1 of Cappa et al., 2019, in which the laboratory and model trends are less complex). We would therefore not go as far as trying to interpret detailed features of the field experiment, since already we fail to reproduce the first-order trends with reasonable parameter values.

Regarding the porosity, we now mention in lines 159-161 that:

*"In a laboratory setting, the sensitivity of the modelled slip rate falls well within the measurement resolution of the sample porosity (typically of the order of several percent of units of porosity), so verification of this sensitivity would be challenging."*

**[R1.6]** *The main observations made here is that the values of H allowing to fit the experimental data recorded during the field injection are strongly larger than the upper bound definition proposed by Niemeijer and Spiers. The model proposed here explain dilation by shear at the contacts of gouge grains. However, along fault interface, dilation is expected to be mostly controlled by fault geometry and long scale roughness encountered within the slip domains. It is stated in the text lines 166-174 but do author think about an adaptation of the model to include a second dilation angle due to fault geometry?*

The idea of a second dilatation angle (or more generally an external contribution to $d\phi/dt$) is an interesting one. We included this suggestion for future work in lines 181-184:

*"For simple, spatially uniform relationships between geometric fault opening and fault slip, this first-order contribution to the fault dilatation may be incorporated into Eq. (1b). However, for more realistic (i.e. spatially heterogeneous) fault opening, a multi-scale numerical extension of the adopted model is required."*

We further note that the CNS model only considers a representative volume element of the size of a few grains, so that incorporation of the long-range fault geometry into the analytical equations demands a careful consideration of the various scales.

**[R1.7]** *Finally, this model is expected to describe the slip behavior of the fault in drained conditions (homogeneous fluid pressure distribution), however, I believe it is not the case in the fluid injection experiments where the slip front outgrowths the fluid pressure front (Bhattacharya and Viesca). In addition, a fluid pressure gradient is expected, even at the scale of the laboratory in partially drained conditions (Passelegue et al., 2018) maybe authors should add a small comment about it in the manuscript.*

We understand the reviewer's concerns, which we considered it at an early stage of this study. For the laboratory experiment, the finite fluid flux indeed demands a fluid pressure gradient within the sample. However, since the region of active deformation is narrow (estimated to be around 60 μm), the characteristic time scale for diffusion is short (less than one μsec at most), and is much less than the characteristic time scale of deformation (the reciprocal of strain rate, being of the order of seconds to minutes). Even for fluid diffusion across the entire gouge layer, the characteristic diffusion time scale is insignificant compared to the time scale of deformation, suggesting that the laboratory experiments can be considered to be fully drained.

For the field experiment, we have two length scales to consider: the across-fault length scale is similar to that of the laboratory experiment, so the volumetric deformation can be taken to occur under drained conditions. The reviewer is correct that in the fault-parallel direction, the fault slip front outpaces the fluid diffusion front. However, the measurements of fault slip are made at the injection point, and so we assumed our single-degree-of-freedom microphysical model to describe the behaviour at this point, not considering long-range interactions. This simplifying assumption then reduces the system to the scales and dimensions of the laboratory sample, for which it was concluded that the deformation conditions are drained.

Minor comments:

**[R1.8]** *Equation 1b: I am probably wrong but I am not sure that the equation described here refer directly to d_phi/d_t. It looks to me more related to the dilatancy rate of the gouge layer.*

Dilatancy (volumetric strain) and porosity are related through: $d\varepsilon = -d\phi/(1-\phi)$. The volumetric strain induced by granular flow is controlled by the dilatancy angle, connecting changes in porosity to fault dilatancy and shear strain.

**[R1.9]** *Lines 98: It is probably not changing a lot, but I wonder if you should not compute Delay_L/L=(phi-phi_0)/(1-/phi_0) to consider the initial fraction of the matrix.*

As was also pointed out by the other reviewer, we neglect variations in $L$ in order to integrate Eq. (5). Consequently, we assume $L \approx L_0$, so that $\Delta L/L \approx (\phi - \phi_0)/(1 - \phi)$. We comment on this in lines 98-102 of the revised manuscript:

"*While we recognise that $L$ varies with $\phi$, integration of (5) does not yield an analytical solution when taking $L = f(\phi)$. Fortunately, as will be shown later, we find that the inferred variations in $L$ are of the order of 10-20% of the absolute value of L, warranting a first-order approximation of a constant value of L. By integrating the above relation from the initial porosity $\phi_0$ up to $\phi$ (c.f. van den Ende et al., 2018), and recognising that $\Delta L/L = (\phi - \phi_0)/(1 - \phi)$ (for $L \approx L_0$), we obtain an expression for the dilatancy $\Delta L$ as a function of slip $\delta$*"

**[R1.10]** *Maybe I missed it but I do not see the definition of $\tilde{a}$ the text, that should be of the form dmu/d(ln V) or something like that.*

In lines 75-76 of the original manuscript, $\tilde{a}$ is defined as "*a proportionality constant for the logarithmic velocity dependence of the grain boundary friction $\tilde{\mu}$*".

**[R1.11]** *Part 4.1: I wonder about the relevance of this part here since the limitation of RSF has been already mentioned in the introduction.*

The limitations of RSF were briefly mentioned in the introduction. Since the CNS model cannot yet be considered to be well-established in the community, and to gently prepare the reader for Section 4.2, we prefer to maintain the comparison between RSF and CNS given by Section 4.1.

---

## Author Comment (AC2) · 8 Oct 2020

**[R2.1]** *In passing from (5) to (6) it seems implicitly assumed that L in the RHS of (5) is constant when integrating, whereas the resulting expression (6) implies that a substantial evolution of L is possible. Could the problem be closed by presenting an equation such as dL/dphi = f(L, phi)?*

Unfortunately, when taking variations in $L$ with $\phi$ into account, the integration of (5) does not yield an analytical solution for the porosity (or $\Delta L$). We comment on this, and justify an assumed constant value of $L$ in lines 98-102 of the revised manuscript:

"*While we recognise that L varies with $\phi$, integration of (5) does not yield an analytical solution when taking $L = f(\phi)$. Fortunately, as will be shown later, we find that the inferred variations in L are of the order of 10-20% of the absolute value of L, warranting a first-order approximation of a constant value of L. By integrating the above relation from the initial porosity $\phi_0$ up to $\phi$ (c.f. van den Ende et al., 2018), and recognising that $\Delta L/L = (\phi - \phi_0)/(1 - \phi)$ (for $L \approx L_0$), we obtain an expression for the dilatancy $\Delta L$ as a function of slip $\delta$*"

**[R2.2]** *Equation (11): wouldn't the linear stability analysis results of Ruina, in which (b-a) takes the place of (b) in (11), provide a more relevant critical stiffness close to steady state?*

In this section, we evaluate the criterion for unstable slip, which may be either seismic or aseismic (e.g. in the form of a slow slip transient). The Ruina-criterion (in terms of (b-a)) provides a criterion for a seismic slip instability, which is harder to derive and thereby less illustrative for the purpose of this section. Moreover, since we did not include a time-dependent compaction mechanism in our simplified CNS formulations, the model system is unable to attain steady-state, which precludes a derivation following the approach of Ruina. For a detailed analysis of the stability of a CNS system, we now include a reference to the work of Chen & Niemeijer (2017) in lines 259-260.

**[R2.3]** *Could the authors flesh out more directly why there is such a strong dependence on the initial porosity (e.g., Fig. 4)?*

In the original manuscript, it was already mentioned in lines 150-152 that the exponential sensitivity of the slip rate to porosity leads to critical behaviour. We've expanded on this by stating (lines 155-156 of the revised manuscript):

"*Since the rate of increase in porosity is proportional to the shear strain rate, which in turn is an exponential function of porosity (refer to Eq. (1b) and (3)), the positive feedback loop leads to an extremely rapidly diverging state.*"

**[R2.4]** *Line 94, typo: "an"*

Corrected